# MIRROR DESCENT POLICY OPTIMIZATION

**Manan Tomar** *
University of Alberta, Amii
manan.tomar@gmail.com

**Lior Shani**
Technion, Israel
shanlior@gmail.com

**Yonathan Efroni**
Microsoft Research NYC
yefroni@microsoft.com

**Mohammad Ghavamzadeh**
Google Research
ghavamza@google.com

## ABSTRACT

Mirror descent (MD), a well-known first-order method in constrained convex optimization, has recently been shown as an important tool to analyze trust-region algorithms in reinforcement learning (RL). However, there remains a considerable gap between such theoretically analyzed algorithms and the ones used in practice. Inspired by this, we propose an efficient RL algorithm, called *mirror descent policy optimization* (MDPO). MDPO iteratively updates the policy by *approximately* solving a trust-region problem, whose objective function consists of two terms: a linearization of the standard RL objective and a proximity term that restricts two consecutive policies to be close to each other. Each update performs this approximation by taking multiple gradient steps on this objective function. We derive *on-policy* and *off-policy* variants of MDPO, while emphasizing important design choices motivated by the existing theory of MD in RL. We highlight the connections between on-policy MDPO and two popular trust-region RL algorithms: TRPO and PPO, and show that explicitly enforcing the trust-region constraint is in fact *not* a necessity for high performance gains in TRPO. We then show how the popular soft actor-critic (SAC) algorithm can be derived by slight modifications of off-policy MDPO. Overall, MDPO is derived from the MD principles, offers a unified approach to viewing a number of popular RL algorithms, and performs better than or on-par with TRPO, PPO, and SAC in a number of continuous and discrete control tasks.

## 1 INTRODUCTION

An important class of RL algorithms consider an additional objective in their policy optimization that aims at constraining the consecutive policies to remain close to each other. These algorithms are referred to as *trust region* or *proximity-based*, resonating the fact that they make the new policy to lie within a trust-region around the old one. This class includes the theoretically grounded conservative policy iteration (CPI) algorithm [15], as well as the state-of-the-art deep RL algorithms, such as trust-region policy optimization (TRPO) [26] and proximal policy optimization (PPO) [28]. The main difference between these algorithms is in the way that they enforce the trust-region constraint. TRPO enforces it explicitly through a line-search procedure that ensures the new policy is selected such that its KL-divergence with the old policy is below a certain threshold. PPO takes a more relaxed approach and updates its policies by solving an unconstrained optimization problem in which the ratio of the new to old policies is clipped to remain bounded. It has been shown that this procedure does not prevent the policy ratios to go out of bound, and only reduces its probability [31, 9].

Mirror descent (MD) [6, 4] is a first-order optimization method for solving constrained convex problems. Although MD is theoretically well-understood in optimization [3, 14], only recently, has it been investigated for policy optimization in RL [25, 12, 20, 29, 1]. Despite the progress made by these results in establishing connections between MD and trust-region policy optimization, there are still considerable gaps between the trust-region RL algorithms that have been theoretically analyzed in their *tabular* form [29] and those that are used in practice, such as TRPO and PPO.

In this paper, motivated by the theory of MD in *tabular* RL, our goal is to derive scaleable and practical RL algorithms from the MD principles, and to use the MD theory to better understand and

---

*Work done partially at FAIR as part of AI Residency.

explain the popular trust-region policy optimization methods. Going beyond the tabular case, when the policy belongs to a parametric class, the trust-region problems for policy update in RL cannot be solved in closed-form. We propose an algorithm, called **mirror descent policy optimization** (MDPO), that addresses this issue by *approximately* solving these trust-region problems via taking **multiple gradient steps** on their objective functions. We derive *on-policy* and *off-policy* variants of MDPO (Section 4). We highlight the connection between on-policy MDPO and TRPO and PPO (Section 4.1), and empirically compare it against these algorithms on several continuous control tasks from OpenAI Gym [7] (Section 5.3). We then show that if we define the trust-region w.r.t. the *uniform policy*, instead of the old one, our off-policy MDPO coincides with the popular soft actor-critic (SAC) algorithm [13]. We discuss this connection in detail (Section 4.2) and empirically compare these algorithms using the same set of continuous control problems (Section 5.4).

Our observations on the comparison between the MDPO algorithms and TRPO, PPO, and SAC are a result of extensive empirical studies on different versions of these algorithms (Section 5 and Appendices E and F). In particular, we first compare the vanilla versions of these algorithms in order to better understand how the core of these methods work relative to each other. We then add a number of code-level optimization techniques derived from the code-bases of TRPO, PPO, and SAC to these algorithms to compare their best form (those that obtain the best results reported in the literature) against each other, while also evaluating MDPO with PPO on 21 Atari games. We address the common belief within the community that explicitly

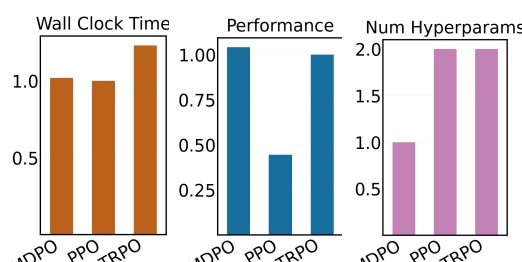

Figure 1: **Overall Comparison**. Between MDPO, PPO, and TRPO, MDPO provides the best trade-off in terms of best average performance, less (normalized) wall clock times, and least number of algorithm specific hyper parameters used.

enforcing the trust-region constraint is a necessity for good performance in TRPO, by showing that MDPO, a trust-region method based on the MD principles, does not require enforcing a hard constraint and achieves strong performance by solely solving an unconstrained problem. We address another common belief that PPO is a better performing algorithm than TRPO. By reporting results of both the vanilla version and the version loaded with code-level optimization techniques for all algorithms, we show that in both cases, TRPO consistently outperforms PPO. This is in line with some of the findings from a recent study on PPO and TRPO [9]. Finally, we provide an optimization perspective for SAC, instead of its initial motivation as an entropy-regularized *(soft)* approximate dynamic programming algorithm. Through comprehensive experiments, we show that on-policy and off-policy MDPO achieve state-of-the-art performance across a number of benchmark tasks, and can be excellent alternatives to popular policy optimization algorithms, such as TRPO, PPO, and SAC.

## 2 PRELIMINARIES

In this paper, we assume that the agent's interaction with the environment is modeled as a $\gamma$-discounted Markov decision process (MDP), denoted by $\mathcal{M} = (\mathcal{S}, \mathcal{A}, P, R, \gamma, \mu)$, where $\mathcal{S}$ and $\mathcal{A}$ are the state and action spaces; $P \equiv P(s'|s, a)$ is the transition kernel; $R \equiv r(s, a)$ is the reward function; $\gamma \in (0, 1)$ is the discount factor; and $\mu$ is the initial state distribution. Let $\pi : \mathcal{S} \to \Delta_{\mathcal{A}}$ be a stationary Markovian policy, where $\Delta_{\mathcal{A}}$ is the set of probability distributions on $\mathcal{A}$. The discounted frequency of visiting a state $s$ by following a policy $\pi$ is defined as $\rho_\pi(s) \equiv (1-\gamma)\mathbb{E}[\sum_{t \geq 0} \gamma^t \mathbb{I}\{s_t = s\}|\mu, \pi]$. The value function of a policy $\pi$ at a state $s \in \mathcal{S}$ is defined as $V^\pi(s) \equiv \mathbb{E}[\sum_{t \geq 0} \gamma^t r(s_t, a_t)|s_0 = s, \pi]$. Similarly, the action-value function of $\pi$ is defined as $Q^\pi(s, a) = \mathbb{E}[\sum_{t \geq 0} \gamma^t r(s_t, a_t)|s_0 = s, a_0 = a, \pi]$. The difference between the action-value $Q$ and value $V$ functions is referred to as the advantage function $A^\pi(s, a) = Q^\pi(s, a) - V^\pi(s)$.

Since finding an optimal policy for an MDP involves solving a non-linear system of equations and the optimal policy may be deterministic (less explorative), many researchers have proposed to add a regularizer in the form of an entropy term to the reward function, and then solve the entropy-regularized (or *soft*) MDP (e.g., [16, 30, 25]). In this formulation, the reward function is modified as $r_\lambda(s, a) = r(s, a) + \lambda H(\pi(\cdot|s))$, where $\lambda$ is the regularization parameter and $H$ is an entropy-related term, such as Shannon entropy [10, 23], Tsallis entropy [17, 24], or relative entropy [2, 22]. Setting

$\lambda = 0$, we return to the original formulation, also referred to as the *hard* MDP. In what follows, we use the terms 'regularized' and 'soft' interchangeably.

## 2.1 MIRROR DESCENT IN CONVEX OPTIMIZATION

Mirror Descent (MD) [4] is a first-order trust-region optimization method for solving constrained convex problems, i.e., $x^* \in \arg\min_{x \in C} f(x)$, where $f$ is a convex function and the constraint set $C$ is convex compact. In each iteration, MD minimizes a sum of two terms: **1)** a linear approximation of the objective function $f$ at the previous estimate $x_k$, and **2)** a proximity term that measures the distance between the updated $x_{k+1}$ and current $x_k$ estimates. MD is considered a trust-region method, since the proximity term keeps the updates $x_k$ and $x_{k+1}$ close to each other. We may write the MD update as

$$x_{k+1} \in \arg\min_{x \in C} \langle \nabla f(x_k), x - x_k \rangle + \frac{1}{t_k} B_\psi(x, x_k), \tag{1}$$

where $B_\psi(x, x_k) := \psi(x) - \psi(x_k) - \langle \nabla \psi(x_k), x - x_k \rangle$ is the Bregman divergence associated with a strongly convex *potential* function $\psi$, and $t_k$ is a step-size determined by the MD analysis. When $\psi = \frac{1}{2}\|\cdot\|_2^2$, the Bergman divergence is the Euclidean distance $B_\psi(x, x_k) = \frac{1}{2}\|x - x_k\|_2^2$, and (1) becomes the *projected gradient descent* algorithm [3]. When $\psi$ is the negative Shannon entropy, the Bregman divergence term takes the form of the KL divergence, i.e., $B_\psi(x, x_k) = \mathrm{KL}(x, x_k)$. In this case, when the constraint set $C$ is the unit simplex, $C = \Delta_\mathcal{X}$, MD becomes the *exponentiated gradient descent* algorithm and (1) has the following closed form [4]:

$$x_{k+1}^i = \frac{x_k^i \exp\left(-t_k \nabla_i f(x_k)\right)}{\sum_{j=1}^n x_k^j \exp\left(-t_k \nabla_j f(x_k)\right)}, \tag{2}$$

where $x_k^i$ and $\nabla_i f$ are the $i^{\text{th}}$ coordinates of $x_k$ and $\nabla f$.

## 3 MIRROR DESCENT IN RL

The goal in RL is to find an optimal policy $\pi^*$. Two common notions of optimality, and as a result, two distinct ways to formulate RL as an optimization problem are as follows:

$$\textbf{(a)} \quad \pi^*(\cdot|s) \in \arg\max_\pi V^\pi(s), \;\; \forall s \in \mathcal{S}, \qquad \textbf{(b)} \quad \pi^* \in \arg\max_\pi \mathbb{E}_{s \sim \mu}\big[V^\pi(s)\big]. \tag{3}$$

In (3a), the value function is optimized over the entire state space $\mathcal{S}$. This formulation is mainly used in value function based RL algorithms. On the other hand, the formulation in (3b) is more common in policy optimization, where a scalar that is the value function at the initial state ($s \sim \mu$) is optimized.

Unlike the MD optimization problem, the objective function is not convex in $\pi$ in either of the above two RL optimization problems. Despite this issue, [12] and [29] have shown that we can still use the general MD update rule (1) and derive MD-style RL algorithms with the update rules

$$\pi_{k+1}(\cdot|s) \leftarrow \arg\max_{\pi \in \Pi} \mathbb{E}_{a \sim \pi}\big[A^{\pi_k}(s, a)\big] - \frac{1}{t_k}\mathrm{KL}(s; \pi, \pi_k), \quad \forall s \in \mathcal{S}, \tag{4}$$

$$\pi_{k+1} \leftarrow \arg\max_{\pi \in \Pi} \mathbb{E}_{s \sim \rho_{\pi_k}}\Big[\mathbb{E}_{a \sim \pi}\big[A^{\pi_k}(s, a)\big] - \frac{1}{t_k}\mathrm{KL}(s; \pi, \pi_k)\Big], \tag{5}$$

for the optimization problems (3a) and (3b), respectively. Note that while in (4), the policy is optimized uniformly over the state space $\mathcal{S}$, in (5), it is optimized over the measure $\rho_{\pi_k}$, i.e., the state frequency induced by the current policy $\pi_k$.

## 4 MIRROR DESCENT POLICY OPTIMIZATION

In this section, we derive *on-policy* and *off-policy* RL algorithms based on the MD-style update rules (4) and (5). We refer to our algorithms as *mirror descent policy optimization* (MDPO). Since the trust-region optimization problems in the update rules (4) and (5) cannot be solved in closed-form, we approximate these updates with multiple steps of stochastic gradient descent (SGD) on the objective functions of these optimization problems. In our *on-policy* MDPO algorithm, described in Section 4.1, we use the update rule (5) and compute the SGD updates using the Monte-Carlo (MC) estimate of the advantage function $A^{\pi_k}$ gathered by following the current policy $\pi_k$. On the other hand, our

*off-policy* MDPO algorithm, described in Section 4.2, is based on the update rule (4) and calculates the SGD update by estimating $A^{\pi_k}$ using samples from a replay buffer.

In our MDPO algorithms, we define the policy space, $\Pi$, as a class of smoothly parameterized stochastic polices, i.e., $\Pi = \{\pi(\cdot|s;\theta) : s \in \mathcal{S}, \theta \in \Theta\}$. We refer to $\theta$ as the policy parameter. We will use $\pi$ and $\theta$ to represent a policy, and $\Pi$ and $\Theta$ to represent the policy space, interchangeably.

## 4.1 ON-POLICY MDPO

In this section, we derive an on-policy RL algorithm based on the MD-based update rule (5), whose pseudo-code is shown in Algorithm 1 in Appendix A. We refer to this algorithm as *on-policy* MDPO. We may write the update rule (5) for the policy space $\Theta$ (defined above) as

$$\theta_{k+1} \leftarrow \arg\max_{\theta \in \Theta} \Psi(\theta, \theta_k), \quad \text{where} \quad \Psi(\theta, \theta_k) = \mathbb{E}_{s \sim \rho_{\theta_k}} \left[ \mathbb{E}_{a \sim \pi_\theta} \left[ A^{\theta_k}(s, a) \right] - \frac{1}{t_k} \text{KL}(s; \pi_\theta, \pi_{\theta_k}) \right]. \quad (6)$$

Each policy update in (6) requires solving a constrained (over $\Theta$) optimization problem. In on-policy MDPO, instead of solving this problem, we update the policy by performing multiple SGD steps on the objective function $\Psi(\theta, \theta_k)$. Interestingly, performing only a single SGD step on $\Psi(\theta, \theta_k)$ is not sufficient as $\nabla_\theta \text{KL}(\cdot; \pi_\theta, \pi_{\theta_k})|_{\theta=\theta_k} = 0$, and thus, if we perform a single-step SGD, i.e.,

$$\nabla_\theta \Psi(\theta, \theta_k)|_{\theta=\theta_k} = \mathbb{E}_{s \sim \rho_{\theta_k} \atop a \sim \pi_\theta} \left[ \nabla \log \pi_{\theta_k}(a|s) A^{\theta_k}(s, a) \right],$$

the resulting algorithm would be equivalent to *vanilla policy gradient* and misses the entire purpose of enforcing the trust-region constraint. As a result, the policy update at each iteration $k$ of on-policy MDPO involves $m$ SGD steps as

$$\theta_k^{(0)} = \theta_k, \quad \text{for} \quad i = 0, \dots, m-1, \quad \theta_k^{(i+1)} \leftarrow \theta_k^{(i)} + \eta \nabla_\theta \Psi(\theta, \theta_k)|_{\theta=\theta_k^{(i)}}, \quad \theta_{k+1} = \theta_k^{(m)},$$

where the gradient of the objective function

$$\nabla_\theta \Psi(\theta, \theta_k)|_{\theta=\theta_k^{(i)}} = \mathbb{E}_{s \sim \rho_{\theta_k} \atop a \sim \pi_{\theta_k}} \left[ \frac{\pi_{\theta_k}^{(i)}}{\pi_{\theta_k}} \nabla \log \pi_{\theta_k^{(i)}}(a|s) A^{\theta_k}(s, a) \right] - \frac{1}{t_k} \mathbb{E}_{s \sim \rho_{\theta_k}} \left[ \nabla_\theta \text{KL}(s; \pi_\theta, \pi_{\theta_k})|_{\theta=\theta_k^{(i)}} \right] \quad (7)$$

can be estimated in an *on-policy* fashion using the data generated by the current policy $\pi_{\theta_k}$. Since in practice, the policy space is often selected as Gaussian, we use the closed-form of KL in this estimation. Our on-policy MDPO algorithm (Algorithm 1, Appendix A) has close connections to two popular on-policy trust-region RL algorithms: TRPO [26] and PPO [28]. We now discuss the similarities and differences between on-policy MDPO and these algorithms.

**Comparison with TRPO**. At each iteration $k$, TRPO considers the constrained optimization problem

$$\max_{\theta \in \Theta} \mathbb{E}_{s \sim \rho_{\theta_k} \atop a \sim \pi_{\theta_k}} \left[ \frac{\pi_\theta(a|s)}{\pi_{\theta_k}(a|s)} A^{\theta_k}(s, a) \right], \quad \text{s.t.} \quad \mathbb{E}_{s \sim \rho_{\theta_k}} \left[ \text{KL}(s; \pi_{\theta_k}, \pi_\theta) \right] \leq \delta, \quad (8)$$

and updates its policy parameter by taking a step in the direction of the *natural gradient* of the objective function in (8) as $\theta_{k+1} \leftarrow \theta_k + \eta F^{-1} \mathbb{E}_{s \sim \rho_{\theta_k} \atop a \sim \pi_{\theta_k}} \left[ \nabla \log \pi_{\theta_k}(a|s) A^{\theta_k}(s, a) \right]$, where $F = \mathbb{E}_{s \sim \rho_{\theta_k} \atop a \sim \pi_{\theta_k}} \left[ \nabla \log \pi_{\theta_k}(a|s) \nabla \log \pi_{\theta_k}(a|s)^\top \right]$ is the *Fisher information matrix* for the current policy $\pi_{\theta_k}$. It then explicitly enforces the trust-region constraint in (8) by a *line-search*: computing the KL-term for $\theta = \theta_{k+1}$ and checking if it is larger than the threshold $\delta$, in which case, the step size is reduced until the constraint is satisfied.

In comparison to TRPO, **first,** on-policy MDPO does not explicitly enforce the trust-region constraint, but approximately satisfies it by performing multiple steps of SGD on the objective function of the optimization problem in the MD-style update rule (6). We say "*it approximately satisfies the constraint*" because instead of fully solving (6), it takes multiple steps in the direction of the gradient of its objective function. **Second,** on-policy MDPO uses simple SGD instead of natural gradient, and thus, does not have to deal with the computational overhead of computing (or approximating) the inverse of the Fisher information matrix.[1] **Third,** the direction of KL in on-policy MDPO, $\text{KL}(\pi, \pi_k)$,

---

[1]TRPO does not explicitly invert $F$, but instead, approximates the natural gradient update using conjugate gradient descent.

is consistent with that in the MD update rule in convex optimization and is different than that in TRPO, $\text{KL}(\pi_k, \pi)$. This does not cause any sampling problem for either algorithm, as both calculate the KL-term in closed-form (Gaussian policies). **Fourth,** while TRPO uses heuristics to define the step-size and to reduce it in case the trust-region constraint is violated, on-policy MDPO uses a simple schedule, motivated by the theory of MD [4], and sets $t_k = 1 - k/K$, where $K$ is the maximum number of iterations. This way it anneals the step-size $t_k$ from 1 to 0 over the iterations of the algorithm.

**Comparison with PPO**. At each iteration $k$, PPO performs multiple steps of SGD on the objective function of the following unconstrained optimization problem:

$$\max_{\theta \in \Theta} \mathbb{E}_{\substack{s \sim \rho_{\theta_k} \\ a \sim \pi_{\theta_k}}} \left[ \min \left\{ \frac{\pi_\theta(a|s)}{\pi_{\theta_k}(a|s)} A^{\theta_k}(s,a), \text{clip}\left( \frac{\pi_\theta(a|s)}{\pi_{\theta_k}(a|s)}, 1 - \epsilon, 1 + \epsilon \right) A^{\theta_k}(s,a) \right\} \right], \quad (9)$$

in which the hyper-parameter $\epsilon$ determines how the policy ratio, $\pi_\theta / \pi_{\theta_k}$, is clipped. It is easy to see that the gradient of the objective function in (9) is zero for the state-action pairs at which the policy ratio is clipped and is non-zero, otherwise. However, since the gradient is averaged over all the state-action pairs in the batch, the policy is updated even if its ratio is out of bound for some state-action pairs. This phenomenon, which has been reported in [31] and [9], shows that clipping in PPO does not prevent the policy ratios to go out of bound, but it only reduces its probability. This means that despite using clipping, PPO does not guarantee that the trust-region constraint is always satisfied. In fact, recent results, including those in [9] and our experiments in Section 5.3, show that most of the improved performance exhibited by PPO is due to code-level optimization techniques, such as learning rate annealing, observation and reward normalization, and in particular, the use of generalized advantage estimation (GAE) [27]. Although both on-policy MDPO and PPO take multiple SGD steps on the objective function of unconstrained optimization problems (6) and (9), respectively, the way they handle the trust-region constraint is completely different.

Another interesting observation is that the adaptive and fixed KL algorithms (we refer to as KL-PPO here), proposed in the PPO paper [28], have policy update rules similar to on-policy MDPO. However, these algorithms have not been used much in practice, because it was shown in the same paper that they perform much worse than PPO. Despite the similarities, there are three main differences between the update rules of KL-PPO and on-policy MDPO. **First,** KL-PPO uses mini-batches whereas MDPO uses the entire data for their multiple ($m$) gradient updates at each round. **Second,** the scheduling scheme used for the $t_k$ parameter is quite different in KL-PPO and MDPO. In particular, KL-PPO either uses a fixed $t_k$ or defines an adaptive scheme that updates (increase/decrease) $t_k$ based on the KL divergence magnitude at that time step. On the other hand, on-policy MDPO uses an annealed schedule to update $t_k$, starting from 1 and slowly bringing it down to near 0. **Third,** similar to TRPO, the direction of KL in KL-PPO, $\text{KL}(\pi_k, \pi)$, is different than that in on-policy MDPO, $\text{KL}(\pi, \pi_k)$. Since in our experiments, on-policy MDPO performs significantly better than PPO (see Section 5.3), we conjecture that either any or a combination of the above differences, especially the first two, is the reason for the inferior performance of KL-PPO, compared to PPO, as reported in [28].

## 4.2 OFF-POLICY MDPO

In this section, we derive an off-policy RL algorithm based on the MD update rule (4). We refer to this as *off-policy* MDPO and provide the pseudo-code in Algorithm 2 in Appendix A. To emulate the uniform sampling over the state space required by (4), Algorithm 2 samples a batch of states from a replay buffer $\mathcal{D}$ (Line 4). While this sampling scheme is not truly uniform, it makes the update less dependent on the current policy. Similar to the on-policy case, we write the update rule (4) for the policy class $\Theta$ as

$$\theta_{k+1} \leftarrow \arg\max_{\theta \in \Theta} \Psi(\theta, \theta_k), \quad \text{where} \quad \Psi(\theta, \theta_k) = \mathbb{E}_{s \sim \mathcal{D}} \left[ \mathbb{E}_{a \sim \pi_\theta} \left[ A^{\theta_k}(s,a) \right] - \frac{1}{t_k} \text{KL}(s; \pi_\theta, \pi_{\theta_k}) \right]. \quad (10)$$

The main idea in Algorithm 2 is to estimate the advantage or action-value function of the current policy, $A^{\theta_k}$ or $Q^{\theta_k}$, in an off-policy fashion, using a batch of data randomly sampled from the replay buffer $\mathcal{D}$. In a similar manner to the policy update of our on-policy MDPO algorithm (Algorithm 1), described in Section 4.1, we then update the policy by taking multiple SGD steps on the objective function $\Psi(\theta, \theta_k)$ of the optimization problem (10) (by keeping $\theta_k$ fixed). A more presentable form of the policy loss in $\Psi$ can be written as follows:

$$L(\theta, \theta_k) = \mathbb{E}_{\substack{s \sim \mathcal{D} \\ \epsilon \sim \mathcal{N}}} \left[ \log \pi_\theta \left( \widetilde{a}_\theta(\epsilon, s) | s \right) - \log \pi_{\theta_k} \left( \widetilde{a}_\theta(\epsilon, s) | s \right) - t_k Q_\psi^{\theta_k} \left( s, \widetilde{a}_\theta(\epsilon, s) \right) \right], \quad (11)$$

In particular, the first two terms here are obtained just by opening the KL, whereas the advantage estimate is replaced by a neural network estimate $Q_\psi$, which is learned from off-policy data in a TD(0) fashion. Furthermore, solely as an implementation detail, another neural network $V_\phi$ is used in conjunction with $Q_\psi$, which is fit to the $Q_\psi$ estimate of the current policy. Finally, the policy loss also uses the reparameterization trick where $\widetilde{a}_\theta(\epsilon, s)$ is the action generated by sampling the $\epsilon$ noise from a zero-mean normal distribution $\mathcal{N}$.

We can easily modify Algorithm 2 to optimize *soft* (entropy regularized) MDPs. In this case, in the critic update (Line 12 of Algorithm 2), the $Q_\psi$ update remains unchanged, while in the $V_\phi$ update, the target changes from $\mathbb{E}_{a \sim \pi_{\theta_{k+1}}}[Q_\psi(\cdot, a)]$ to $\mathbb{E}_{a \sim \pi_{\theta_{k+1}}}[Q_\psi(\cdot, a) - \lambda \log \pi(a|\cdot)]$. The loss function (11) used for the actor (policy) update (Lines 7-9 of Algorithm 2) is also modified, $Q_\psi$ becomes the soft $Q$-function and a term $\lambda t_k \log \pi_{\theta_k}(\widetilde{a}_\theta(\epsilon, s)|s)$ is added inside the expectation. We denote these changes explicitly in Algorithm 3.

Similarly to on-policy MDPO that has close connection to TRPO and PPO, discussed in Section 4.1, off-policy MDPO (Algorithm 2 and 3) is related to the popular soft actor-critic (SAC) algorithm [13]. We now derive SAC by slight modifications in the derivation of off-policy MDPO. This gives an optimization interpretation to SAC, which we then use to show strong ties between the two algorithms.

**Comparison with SAC**. Soft actor-critic is an approximate policy iteration algorithm in soft MDPs. At each iteration $k$, it first estimates the (soft) $Q$-function of the current policy, $Q^{\pi_k}$, and then sets the next policy to the (soft) greedy policy w.r.t. the estimated $Q$-function as

$$\pi_{k+1}(a|s) \leftarrow \exp\left(Q^{\pi_k}(s, a)\right) / Z^{\text{SAC}}(s), \tag{12}$$

where $Z^{\text{SAC}}(s) = \mathbb{E}_{a \sim \pi_k(\cdot|s)}\left[\exp\left(Q^{\pi_k}(s, a)\right)\right]$ is a normalization term. However, since tractable policies are preferred in practice, SAC suggests to project the improved policy back into the policy space considered by the algorithm, using the following optimization problem:

$$\theta_{k+1} \leftarrow \operatorname*{arg\,min}_{\theta \in \Theta} \mathcal{L}^{\text{SAC}}(\theta, \theta_k), \qquad \mathcal{L}^{\text{SAC}}(\theta, \theta_k) = \mathbb{E}_{s \sim \mathcal{D}}\left[\text{KL}\left(s; \pi_\theta, \frac{\exp\left(Q^{\theta_k}(s, \cdot)\right)}{Z^{\text{SAC}}(s)}\right)\right]. \tag{13}$$

This update rule computes the next policy as the one with the minimum KL-divergence to the term on the RHS of (12)

Since the optimization problem in (13) is invariant to the normalization term, unlike (12), the policy update (13) does not need to compute $Z^{\text{SAC}}(s)$. By writing the KL definition and using the reparameterization trick in (13), SAC updates its policy by minimizing the following loss function:

$$L^{\text{SAC}}(\theta, \theta_k) = \mathbb{E}_{\substack{s \sim \mathcal{D} \\ \epsilon \sim \mathcal{N}}}\left[\lambda \log \pi_\theta\left(\widetilde{a}_\theta(\epsilon, s)|s\right) - Q_\psi^{\theta_k}\left(s, \widetilde{a}_\theta(\epsilon, s)\right)\right]. \tag{14}$$

Comparing the loss in (14) with the one used in off-policy MDPO (Eq. 11), we notice that despite the similarities, the main difference is the absence of the current policy, $\pi_{\theta_k}$, in the SAC loss function. To explain the relationship between off-policy MDPO and SAC, recall from Section 2.1 that if the constraint set is the unit simplex, i.e., $C = \Delta_{\mathcal{X}}$, the MD update has the closed-form shown in (2). Thus, if the policy class (constraint set) $\Pi$ in the update rule (4) is the entire space of stochastic policies, then we may write (4) in closed-form as (see e.g., [21, 29])

$$\pi_{k+1}(a|s) \leftarrow \pi_k(a|s) \exp\left(t_k Q^{\pi_k}(s, a)\right) / Z(s), \tag{15}$$

where $Z(s) = \mathbb{E}_{a \sim \pi_k(\cdot|s)}\left[\exp\left(t_k Q^{\pi_k}(s, a)\right)\right]$ is a normalization term. The closed-form solution (15) is equivalent to solving the constrained optimization problem (4) in two phases (see [14]): **1)** solving the *unconstrained* version of (4) that leads to the numerator of (15), followed by **2)** *projecting* this (unconstrained) solution back into the constrained set (all stochastic policies) using the same choice of Bregman divergence (KL in our case), which accounts for the normalization term in (15). Hence, when we optimize over the parameterized policy space $\Theta$ (instead of all stochastic policies), the MD update would be equivalent to finding a policy $\theta \in \Theta$ with minimum KL-divergence to the solution of the *unconstrained* optimization problem obtained in the first phase (the numerator of Eq. 15). This leads to the following policy update rule:

$$\theta_{k+1} \leftarrow \operatorname*{arg\,min}_{\theta \in \Theta} \mathcal{L}(\theta, \theta_k), \qquad \mathcal{L}(\theta, \theta_k) = \mathbb{E}_{s \sim \mathcal{D}}\left[\text{KL}\left(s; \pi_\theta, \pi_{\theta_k} \exp(t_k Q^{\theta_k})\right)\right]. \tag{16}$$

If we write the definition of KL and use the reparameterization trick in (16), we will rederive the loss function (11) used by our off-policy MDPO algorithm.[2] Note that both SAC (13) and off-policy MDPO (16) use KL projection to project back to the set of policies. For SAC, the authors argue that any projection can be chosen arbitrarily. However, our derivation clearly shows that the selection of KL projection is dictated by the choice of the Bregman divergence.

As mentioned earlier, the main difference between the loss functions used in the policy updates of SAC (14) and off-policy MDPO (11) is the absence of the current policy, $\pi_{\theta_k}$, in the SAC's loss function.[3] The current policy, $\pi_{\theta_k}$, appears in the policy update of off-policy MDPO, because it is a trust-region algorithm, and thus, tries to keep the new policy close to the old one. On the other hand, following the original interpretation of SAC as an approximate dynamic programming algorithm, its policy update does not contain a term to keep the new and old policies close to each other. It is interesting to note that SAC's loss function can be re-obtained by repeating the derivation which leads to off-policy MDPO, and replacing the current policy, $\pi_{\theta_k}$, with the *uniform policy* in the objective (10) of off-policy MDPO. Therefore, SAC can be considered as a trust-region algorithm w.r.t. the uniform policy (or an entropy regularized algorithm). This means its update encourages the new policy to remain *explorative*, by keeping it close to the uniform policy.

Trust-PCL [22] uses the path consistency idea along with entropy regularization and an additional term for remaining close to a past policy. In principle, this resembles the off-policy MDPO algorithm. However, Trust-PCL uses a multi-step consistency loss whereas off-policy MDPO uses single transitions. Moreover, besides different derivations, there remain implementation-level details between the two, as Trust-PCL only uses a $V$ network while off-policy MDPO uses a $Q$ function as well.

Due to space constraints, we defer a discussion on the forward and reverse KL directions (including the ECPO [21] algorithm) to Appendix D.

## 5 EXPERIMENTAL RESULTS

In this section, we empirically evaluate our on-policy and off-policy MDPO algorithms on a number of continuous control tasks from OpenAI Gym [7], and compare them with state-of-the-art baselines: TRPO, PPO, and SAC. We report all experimental details, including the hyper-parameter values used by the algorithms, in Appendix B. In the tabular results, both in the main paper and in Appendices E and F, we report the final training scores averaged over 5 runs and their 95% confidence intervals (CI). We bold-face the values with the best mean scores. We also compare on-policy MDPO and PPO on 21 Atari games from the ALE benchmark [5], showing averages over 5 random seeds. We strictly follow the hyperparameters reported in the PPO paper, and use $m = 3$ for all games.

For off-policy MDPO, we experiment with two potential functions $\psi$ to define the Bregman divergence $B_\psi$: **1)** Shannon entropy, which results in the KL version (described in Section 4.2), and **2)** Tsallis entropy, which results in the Tsallis version of off-policy MDPO. We refer the reader to Appendix C for the complete description and detailed derivation of the Tsallis version. Note that we did not pursue a similar bifurcation between Tsallis and KL induced Bregman divergences for the on-policy case since the exact derivations are more tedious there. Another important point to note is that the Tsallis entropy gives us a range of entropies, con-

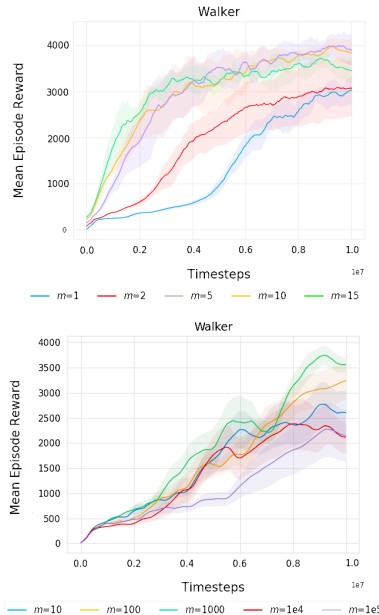

Figure 2: **Performance of on-policy (top) and off-policy (bottom) MDPO** (code level optimizations included) for different values of $m$ on the Walker2d task.

---

[2]In soft MDPs, $Q^{\pi_k}$ is replaced by its soft version and a term $-\lambda t_k \log \pi_{\theta_k}(a|s)$ is added to the exponential in (15). This will result in the same changes in (16). Similar to the hard case, applying the reparameterization trick to the the soft version of (16) gives us the soft version of the loss function (11).

[3]The same difference can also be seen in the policy updates (12) and (15) of SAC and off-policy MDPO.

|  | | On-Policy | | | Off-Policy | |
| Env | MDPO | TRPO | PPO | MDPO-KL | MDPO-Tsallis | SAC |
| --- | --- | --- | --- | --- | --- | --- |
| Hopper-v2 | **2361** ($\pm$ 518) | 1979 ($\pm$ 672) | 2051 ($\pm$ 241) | **2428** ($\pm$ 395) | **2428** ($\pm$ 395), $q = 1.0$ | 1870 ($\pm$ 404) |
| Walker2d-v2 | **4834** ($\pm$ 607) | 4473 ($\pm$ 558) | 1490 ($\pm$ 292) | 3591 ($\pm$ 366) | **4028** ($\pm$ 287), $q = 2.0$ | 3738 ($\pm$ 312) |
| HalfCheetah-v2 | **4172** ($\pm$ 1156) | 3751 ($\pm$ 910) | 2041 ($\pm$ 1319) | 11823 ($\pm$ 154) | 11823 ($\pm$ 154), $q = 1.0$ | **11928** ($\pm$ 342) |
| Ant-v2 | **5211** ($\pm$ 43) | 4682 ($\pm$ 278) | 59 ($\pm$ 133) | 4434 ($\pm$ 749) | **5486** ($\pm$ 737), $q = 2.0$ | 4989 ($\pm$ 579) |
| Humanoid-v2 | 3234 ($\pm$ 566) | **4414** ($\pm$ 132) | 529 ($\pm$ 47) | 5323 ($\pm$ 348) | **5611** ($\pm$ 260), $q = 1.2$ | 5191 ($\pm$ 312) |
| H. Standup-v2 | **155261** ($\pm$ 3898) | 149847 ($\pm$ 2632) | 97223 ($\pm$4479) | 143955 ($\pm$ 4499) | **165882** ($\pm$ 16604), $q = 1.4$ | 154765 ($\pm$ 11721) |

Table 1: **Comparisons on MuJoCo domains**. Averaged (over 5 runs) returns for **Loaded+GAE** version of MDPO, TRPO, PPO, and SAC algorithms, together with their $95\%$ confidence intervals. **On-policy** results are for **10M** timesteps. The values with the best mean scores are bold-faced.

trolled by the parameter $q \in (0, 2]$ (see Appendix C). Two special cases are **1)** Shannon entropy for $q = 1.0$, and **2)** sparse Tsallis for $q = 2.0$ [17, 18, 24].

## 5.1 ON MULTIPLE SGD STEPS

In on-policy MDPO, we implement the multi-step update at each MD iteration of the algorithm, by sampling $M$ trajectories from the current policy, generating estimates of the advantage function, and performing $m$ gradient steps using the same set of trajectories. We evaluated on-policy MDPO for different values of $m$ in all tasks. We show the results for Walker2d in Figure 2 (top). The results for all tasks show a clear trade-off between $m$ and the performance. Moreover, $m = 10$ seems to be the best value across the tasks. This is why we use $m = 10$ in all our on-policy MDPO experiments. Our results clearly indicate that using $m = 1$ leads to inferior performance as compared to $m = 10$, reaffirming the theory that suggests solving the trust-region problem in RL requires taking several gradient steps at each MD iteration. Finally, in our preliminary experiments with TRPO, we observed that performing multiple gradient steps at each iteration of TRPO does not lead to any improvement, sometimes even leading to worse performance than when performing a single-step update.

For off-policy MDPO, performing multiple SGD steps at each MD iteration (Lines 6 to 10 in Algorithm 2) becomes increasingly time-consuming as the value of $m$ grows. This is because off-policy algorithms perform substantially more gradient updates than their on-policy counterparts (a gradient step per environment step vs. a gradient step per almost $1,000$ environment steps). To address this issue, we resort to staying close to an $m$-step old copy of the current policy, while performing a single gradient update at each iteration of the algorithm. This copy is updated every $m$ iterations with the parameters of the current policy. Our results for the Hopper domain in Appendix G.1 show that the performance of MDPO can be improved by performing $m$ gradient updates at each iteration, but we omit from performing these experiments at scale because of their unreasonably high wall-clock time. Finally, we evaluated off-policy MDPO for different values of $m$ in all tasks and show the results for Walker2d in Figure 2 (bottom). We found it hard to identify a single best value of $m$ for all tasks. However, $m = 1000$ had the most reasonable performance across the tasks, and thus, we use it in all our off-policy MDPO experiments.

## 5.2 ON CODE-LEVEL OPTIMIZATIONS

There are certain "code-level optimization techniques" used in code-bases of TRPO, PPO, and SAC that result in enhanced performance. In [9], the authors provided a case study of these techniques in TRPO and PPO. We provide a detailed description of these techniques in Appendix B, and report the performance of the algorithms without these techniques (vanilla or minimal version) and with these techniques (loaded and loaded+GAE versions) in Appendices E and F. Note that the loaded+GAE version of TRPO and PPO match their state-of-the-art results in the literature.

Overall, the key takeaway from our results is that MDPO performs significantly better than PPO and on-par or better than TRPO and SAC, while being much simpler to implement, and more general as being derived from the theory of MD in RL. In the next two sections, we report our main observations from our on-policy and off-policy experiments.

## 5.3 ON-POLICY RESULTS

We implemented three versions of on-policy MDPO, TRPO, and PPO: **1)** the vanilla or minimal version, **2)** the loaded version in which we add the code-level optimization techniques to these

algorithms, and **3)** the loaded version plus GAE, whose results are reported in Table 1. The results for all three versions are reported in Appendix E.

We elicit the following observations from our results. **First,** on-policy MDPO performs better than or on par with TRPO and better than PPO across all tasks. This contradicts the common belief that explicitly enforcing the constraint (e.g., through line-search) as done in TRPO is necessary for achieving good performance. **Second,** on-policy MDPO can be implemented more efficiently than TRPO, because it does not require the extra line-search step. Notably, TRPO suffers from scaling issues as it requires computing the correct step-size of the gradient update using a line search, which presents as an incompatible part of the computation graph in popular auto-diff packages, such as TensorFlow. Moreover, MDPO performs significantly better than PPO, while remaining equally efficient in terms of implementation. **Third,** TRPO performs better than PPO consistently, both in the vanilla case and when the code-level optimizations (including GAE) are added to both algorithms. This is in contrast to the common belief that PPO is a better performing algorithm than TRPO. Our observation is in line with what noted in the empirical study of these two algorithms in [9], and we believe it further reinforces it. Adding code-level optimizations and GAE improve the performance of PPO, but not enough to outperform TRPO, when it also benefits from these additions. Lastly, **fourth,** it was shown in [31] that PPO is prone to instability issues. Our experiments show that this is indeed the case as PPO's performance improves until the standard time-step mark of 1M, and then decreases in some tasks. For example, in the Ant-v2 domain, both PPO and TRPO get to a similar score ( 1000) around the 1M mark but then PPO's performance decreases whereas TRPO continues to increase, as can be seen in Table 1 and Appendix E.

**Atari results**. To show that MDPO can be robustly used as an excellent substitute for PPO, we compare the two algorithms on 21 games from the ALE benchmark. Our results show that MDPO performs better or on par than PPO on 15 out of 21 games, while performing better than PPO on 6 out of 21 games. Due to space constraints, we report the full training plots in Appendix 10. Interestingly, both MDPO and PPO behave quite differently in a lot of games. Since we do not optimize any hyperparameters for MDPO, it might be possible to get more gains with further finetuning. Note that it is well known that TRPO leads to much inferior performance than PPO on the ALE benchmark. Indeed, comparing our results with those in the TRPO paper, we see that both MDPO and PPO win in 5 out of the 6 games reported in the TRPO paper.

## 5.4 Off-policy Results

Similar to the on-policy case, we implemented both vanilla and loaded versions of off-policy MDPO and SAC. We report the results of the loaded version in this section (Table 1), and the complete results in Appendix F. We observe the following from these results. **First,** off-policy MDPO-KL performs on par with SAC across all tasks. **Second,** off-policy MDPO-Tsallis that has an extra hyper-parameter $q$ to tune can outperform SAC across all tasks. We observe that the best performing values of $q$ are different for each domain but always lie in the interval $[1.0, 2.0]$. **Third,** off-policy MDPO results in a performance increase in most tasks, both in terms of sample efficiency and final performance, in comparison to on-policy MDPO. This is consistent with the common belief about the superiority of off-policy to on-policy algorithms.

Similar to off-policy MDPO, we can incorporate the Tsallis entropy in SAC. In [18], the authors showed performance improvement over SAC by properly tuning the value of $q$ in SAC-Tsallis. However, in domains like Humanoid-v2 and Ant-v2, they only reported results for the 1M time-step mark, instead of the standard 3M. In our preliminary experiments with SAC-Tsallis in Appendix G.3, we did not see much improvement over SAC by tuning $q$, unlike what we observed in our MDPO-Tsallis results. More experiments and further investigation are hence needed to better understand the effect of Tsallis entropy (and $q$) in these algorithms.

## 6 Conclusions

We derived on-policy and off-policy algorithms from the theory of MD in RL. Each policy update in our MDPO algorithms is formulated as a trust-region optimization problem. However, our algorithms do not update their policies by solving these problems, instead, update them by taking *multiple gradient steps* on the objective function of these problems. We described in detail the relationship between on-policy MDPO and TRPO and PPO. We also discussed how SAC can be derived by slight modifications of off-policy MDPO. Finally, using a comprehensive set of experiments, we showed

that on-policy and off-policy MDPO can achieve performance better than or equal to these three popular RL algorithms, and thus can be considered as excellent alternatives to them.

We can think of several future directions. In addition to evaluating MDPO algorithms in more complex and realistic problems, we would like to see their performance in discrete action problems in comparison with algorithms like DQN and PPO. Investigating the use of Bregman divergences other than KL seems to be promising. Our work with Tsallis entropy is in this direction but more algorithmic and empirical work needs to be done. Finally, there are recent theoretical results on incorporating exploration into the MD-based updates. Applying exploration to MDPO could prove most beneficial, especially in complex environments.

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

# Appendix

## A PSEUDOCODES

Below we provide the pseudocodes for the two MDPO algorithms, on-policy and off-policy.

---

**Algorithm 1** On-Policy MDPO

---

1: **Initialize** Value network $V_\phi$; Policy networks $\pi_{\text{new}}$ and $\pi_{\text{old}}$;
2: **for** $k = 1, \ldots, K$ **do**
3:     # On-policy Data Generation
4:     Simulate the current policy $\pi_{\theta_k}$ for $M$ steps;
5:     **for** $t = 1, \ldots, M$ **do**
6:         Calculate return $R_t = R(s_t, a_t) = \sum_{j=t}^{M} \gamma^{j-t} r_j$;   Estimate advantage $A(s_t, a_t) = R(s_t, a_t) - V_\phi(s_t)$;
7:     **end for**
8:     # Policy Improvement   *(Actor Update)*
9:     $\theta_k^{(0)} = \theta_k$;
10:     **for** $i = 0, \ldots, m - 1$ **do**
11:         $\theta_k^{(i+1)} \leftarrow \theta_k^{(i)} + \eta \nabla_\theta \Psi(\theta, \theta_k)\big|_{\theta = \theta_k^{(i)}}$;                                                      (Eq. 7)
12:     **end for**
13:     $\theta_{k+1} = \theta_k^{(m)}$;
14:     # Policy Evaluation   *(Critic Update)*
15:     Update $\phi$ by minimizing the $N$-minibatch ($N \leq M$) loss function   $L_{V_\phi} = \frac{1}{N} \sum_{t=1}^{N} \left[ V_\phi(s_t) - R_t \right]^2$;
16: **end for**

---

---

**Algorithm 2** Off-Policy MDPO

---

1: **Initialize** Replay buffer $\mathcal{D} = \emptyset$; Value networks $V_\phi$ and $Q_\psi$; Policy networks $\pi_{\text{new}}$ and $\pi_{\text{old}}$;
2: **for** $k = 1, \ldots, K$ **do**
3:     Take action $a_k \sim \pi_{\theta_k}(\cdot|s_k)$, observe $r_k$ and $s_{k+1}$, and add $(s_k, a_k, r_k, s_{k+1})$ to the replay buffer $\mathcal{D}$;
4:     Sample a batch $\{(s_j, a_j, r_j, s_{j+1})\}_{j=1}^N$ from $\mathcal{D}$;
5:     # Policy Improvement   *(Actor Update)*
6:     $\theta_k^{(0)} = \theta_k$;
7:     **for** $i = 0, \ldots, m-1$ **do**
8:         $\theta_k^{(i+1)} \leftarrow \theta_k^{(i)} + \eta \nabla_\theta L(\theta, \theta_k)|_{\theta = \theta_k^{(i)}}$;                    (Eq. 11)
9:     **end for**
10:     $\theta_{k+1} = \theta_k^{(m)}$;
11:     # Policy Evaluation   *(Critic Update)*
12:     Update $\phi$ and $\psi$ by minimizing the loss functions
        $L_{V_\phi} = \frac{1}{N} \sum_{j=1}^N \left[ V_\phi(s_j) - Q_\psi\big(s_j, \pi_{\theta_{k+1}}(s_j)\big) \right]^2$;
        $L_{Q_\psi} = \frac{1}{N} \sum_{j=1}^N \left[ r(s_j, a_j) + \gamma V_\phi(s_{j+1}) - Q_\psi(s_j, a_j) \right]^2$;
13: **end for**

---

$$L(\theta, \theta_k) = \mathbb{E}_{\substack{s \sim \mathcal{D} \\ \epsilon \sim \mathcal{N}}} \Big[ \log \pi_\theta\big(\widetilde{a}_\theta(\epsilon, s)|s\big) - \log \pi_{\theta_k}\big(\widetilde{a}_\theta(\epsilon, s)|s\big) - t_k Q_\psi^{\theta_k}\big(s, \widetilde{a}_\theta(\epsilon, s)\big) \Big] \quad \text{(Eq. 11 revisited)}$$

---

**Algorithm 3** Off-Policy MDPO (Soft)

---

1: **Initialize** Replay buffer $\mathcal{D} = \emptyset$; Value networks $V_\phi$ and $Q_\psi$; Policy networks $\pi_{\text{new}}$ and $\pi_{\text{old}}$;
2: **for** $k = 1, \ldots, K$ **do**
3:     Take action $a_k \sim \pi_{\theta_k}(\cdot|s_k)$, observe $r_k$ and $s_{k+1}$, and add $(s_k, a_k, r_k, s_{k+1})$ to the replay buffer $\mathcal{D}$;
4:     Sample a batch $\{(s_j, a_j, r_j, s_{j+1})\}_{j=1}^N$ from $\mathcal{D}$;
5:     # Policy Improvement   *(Actor Update)*
6:     $\theta_k^{(0)} = \theta_k$;
7:     **for** $i = 0, \ldots, m-1$ **do**
8:         $\theta_k^{(i+1)} \leftarrow \theta_k^{(i)} + \eta \nabla_\theta L(\theta, \theta_k)|_{\theta = \theta_k^{(i)}}$;                    (Eq. 11 soft)
9:     **end for**
10:     $\theta_{k+1} = \theta_k^{(m)}$;
11:     # Policy Evaluation   *(Critic Update)*
12:     Update $\phi$ and $\psi$ by minimizing the loss functions
        $L_{V_\phi} = \frac{1}{N} \sum_{j=1}^N \left[ V_\phi(s_j) - Q_\psi\big(s_j, \pi_{\theta_{k+1}}(s_j)\big) - \lambda \log \pi_{\theta_{k+1}}(s_j) \right]^2$;
        $L_{Q_\psi} = \frac{1}{N} \sum_{j=1}^N \left[ r(s_j, a_j) + \gamma V_\phi(s_{j+1}) - Q_\psi(s_j, a_j) \right]^2$;
13: **end for**

---

$$L(\theta, \theta_k) = \mathbb{E}_{\substack{s \sim \mathcal{D} \\ \epsilon \sim \mathcal{N}}} \Big[ \log \pi_\theta\big(\widetilde{a}_\theta(\epsilon, s)|s\big) - (1 - \lambda t_k) \log \pi_{\theta_k}\big(\widetilde{a}_\theta(\epsilon, s)|s\big) - t_k Q_\psi^{\theta_k}\big(s, \widetilde{a}_\theta(\epsilon, s)\big) \Big] \quad \text{(Eq. 11 soft)}$$

## B  EXPERIMENTAL DETAILS

### B.1  SETUP

We evaluate all algorithms on OpenAI Gym [7] based continuous control tasks, including Hopper-v2, Walker2d-v2, HalfCheetah-v2, Ant-v2, Humanoid-v2 and HumanoidStandup-v2. All experiments are run across 5 random seeds. Each plot shows the empirical mean of the random runs while the shaded region represents a 95% confidence interval (*empirical mean* $\pm 1.96 \times$ *empirical standard deviation /* $\sqrt{n = 5}$). We report results in both figure and tabular forms. The tabular results denote the mean final training performance and the best values with overlapping confidence intervals are bolded.

For all off-policy experiments, we use $\lambda = 0.2$ across all tasks, which is known to be the best performing value for all tasks according to [13] (In our experiments, a value of 0.2 worked equally well for Humanoid as the reported 0.05 in the SAC paper). We report all details of our off-policy experiments including hyperparameter values in Table 3. Moreover, since doing multiple gradient steps at each iteration becomes quite time consuming for the off-policy case, we get around this issue by fixing the old policy ($\pi_{\theta_k}$) for $m$ number of gradient steps, in order to mimic the effect from taking multiple gradients steps at each iteration. This ensures that the total number of environment steps are always equal to the total number of gradients steps, irrespective of the value of $m$. Finally, for all experiments, we use a fixed Bregman stepsize ($1/t_k$) as opposed to an annealed version like in the on-policy case.

### B.2  CODE-LEVEL OPTIMIZATION TECHNIQUES

The widely available OpenAI Baselines [8] based PPO implementation uses the following five major modifications to the original algorithm presented in [28] – value function clipping, reward normalization, observation normalization, orthogonal weight initialization and an annealed learning rate schedule for the Adam optimizer. These are referred to as code level optimization techniques (as mentioned in above sections) and are originally noted in [9]. Following the original notation, we refer to the vanilla or minimal version of PPO, i.e. without these modifications as PPO-M. Then, we consider two PPO versions which include all such code level optimizations, with the hyperparameters given in [28]. One of them does not use GAE while the other version includes GAE. Therefore they are referred to as PPO-LOADED and PPO-LOADED+GAE respectively. These versions, although being far from the theory, have been shown to be the best performing ones, and so form as a good baseline. We do a similar bifurcation for TRPO and on-policy MDPO. We report all details of our on-policy experiments including hyperparameter values in Table 2.

Similarly, for the off-policy MDPO versions, we again restrain from using the optimization tricks mentioned above. However we do employ three techniques that are common in actor-critic based algorithms, namely: using separate $Q$ and $V$ functions as in [13], using two $Q$ functions to reduce overestimation bias and using soft target updates for the value function. Prior work [11, 19] has shown these techniques help improve stability.

Similar to the on-policy experiments, we include a minimal and loaded version for the off-policy experiments as well, which are described in Appendix D. In particular, this branching is done based on the neural network and batch sizes used. Since the standard values in all on-policy algorithms is different from the standard values used by most off-policy approaches, we show results for both set of values. This elicits a better comparison between on-policy and off-policy methods.

| Hyperparameter | TRPO-M | TRPO-LOADED | PPO-M | PPO-LOADED | MDPO-M | MDPO-LOADED |
|---|---|---|---|---|---|---|
| Adam stepsize | - | - | $3 \times 10^{-4}$ | Annealed from 1 to 0 | $3 \times 10^{-4}$ | Annealed from 1 to 0 |
| minibatch size | 128 | 128 | 64 | 64 | 128 | 128 |
| number of gradient updates ($m$) | - | - | - | - | 5 | 10 |
| reward normalization | ✗ | ✓ | ✗ | ✓ | ✗ | ✓ |
| observation normalization | ✗ | ✓ | ✗ | ✓ | ✗ | ✓ |
| orthogonal weight initialization | ✗ | ✓ | ✗ | ✓ | ✗ | ✓ |
| value function clipping | ✗ | ✓ | ✗ | ✓ | ✗ | ✓ |
| GAE $\lambda$ | 1.0 | 0.95 | 1.0 | 0.95 | 1.0 | 0.95 |
| horizon (T) | | | | 2048 | | |
| entropy coefficient | | | | 0.0 | | |
| discount factor | | | | 0.99 | | |
| total number of timesteps | | | | $10^7$ | | |
| #runs used for plot averages | | | | 5 | | |
| confidence interval for plot runs | | | | $\sim 95\%$ | | |

Table 2: Hyper-parameters of all on-policy methods.

| Hyperparameter | MDPO-M KL | MDPO-M Tsallis | SAC-M | MDPO-LOADED KL | MDPO-LOADED Tsallis | SAC-LOADED |
|---|---|---|---|---|---|---|
| number of hidden units per layer | 64 | 64 | 64 | 256 | 256 | 256 |
| minibatch size | 64 | 64 | 64 | 256 | 256 | 256 |
| entropy coefficient ($\lambda$) | | | | 0.2 | | |
| Adam stepsize | | | | $3 \times 10^{-4}$ | | |
| reward normalization | | | | ✗ | | |
| observation normalization | | | | ✗ | | |
| orthogonal weight initialization | | | | ✗ | | |
| value function clipping | | | | ✗ | | |
| replay buffer size | | | | $10^6$ | | |
| target value function smoothing coefficient | | | | 0.005 | | |
| number of hidden layers | | | | 2 | | |
| discount factor | | | | 0.99 | | |
| #runs used for plot averages | | | | 5 | | |
| confidence interval for plot runs | | | | $\sim 95\%$ | | |

Table 3: Hyper-parameters of all off-policy methods.

| | Hopper-v2 | Walker2d-v2 | HalfCheetah-v2 | Ant-v2 | Humanoid-v2 | HumanoidStandup-v2 |
|---|---|---|---|---|---|---|
| Bregman stepsize ($1/t_k$) | 0.8 | 0.4 | 0.3 | 0.5 | 0.5 | 0.3 |

Table 4: Bregman stepsize for each domain, used by off-policy MDPO.

## C  TSALLIS-BASED BREGMAN DIVERGENCE

As described in section 2.1, the MD update contains a Bregman divergence term. A Bregman divergence is a measure of distance between two points, induced by a strongly convex function $\psi$. In the case where the potential function $\psi$ is the negative Shannon entropy, the resulting Bregman is the KL divergence. Similarly, when $\psi$ is the negative Tsallis entropy, for a real number $q$, i.e.,

$$\psi(\pi) = \frac{1}{1-q}\Big(1 - \sum_a \pi(a \mid s)^q\Big), \tag{17}$$

we obtain the Tsallis Bregamn divergence, i.e.,

$$B_\psi(\pi, \pi_k) = \frac{q}{1-q}\sum_a \pi(a \mid s)\pi_k(a \mid s)^{q-1} - \frac{1}{1-q}\sum_a \pi(a \mid s)^q + \sum_a \pi_k(a \mid s)^q. \tag{18}$$

Note that the last term on the RHS of (18) is independent of the policy $\pi$ being optimized. Also note that as $q \to 1$, the Tsallis entropy collapses to the Shannon entropy $-\sum_a \pi(a \mid s)\log\pi(a \mid s)$, and thus, it generalizes the Shannon entropy. Moreover, for $q = 2$, the Tsallis entropy is called the *sparse* Tsallis entropy.

At first glance, the above expression is very different from the definition of the KL divergence. However, by defining the function $\log_q$ as

$$\log_q x := \begin{cases} \frac{x^{q-1}-1}{q-1}, & \text{if } q \neq 1 \text{ and } x > 0, \\ \log q, & \text{if } q = 1 \text{ and } x > 0, \end{cases}$$

we may write the negative Tsallis entropy, defined by (17), as

$$\psi(\pi) = \sum_a \pi(a \mid s)\log_q \pi(a \mid s), \tag{19}$$

and the Tsallis Bregman, defined by (18), in a similar manner to the KL divergence as

$$B_\psi(\pi, \pi_k) = \underbrace{\sum_a \pi(a \mid s)\big(\log_q \pi(a \mid s) - q\log_q \pi_k(a \mid s)\big)}_{= \text{KL}(\pi, \pi_k), \text{ for } q=1} - \overbrace{\underbrace{(1-q)\sum_a \pi_k(a \mid s)\log_q \pi_k(a \mid s)}_{=0, \text{ for } q=1}}^{\text{independent of the policy } \pi \text{ being optimized}}. \tag{20}$$

With this convenient definition, we can write the Tsallis-based version of the off-policy MDPO objective defined in (11) as

$$L^{\text{Tsallis}}(\theta, \theta_k) = \mathbb{E}_{\substack{s \sim \mathcal{D} \\ \epsilon \sim \mathcal{N}}}\big[\log_q \pi_\theta\big(\widetilde{a}_\theta(\epsilon, s)|s\big) - q\log_q \pi_{\theta_k}\big(\widetilde{a}_\theta(\epsilon, s)|s\big) - t_k Q_\psi^{\theta_k}\big(s, \widetilde{a}_\theta(\epsilon, s)\big)\big]. \tag{21}$$

Note that the last term on the RHS of (20) is independent of the policy being optimized (i.e., $\pi$ or $\theta$), and thus, does not appear in the loss function $L^{\text{Tsallis}}(\theta, \theta_k)$ in (21).

Note that on-policy MDPO uses a closed form version for the Bregman divergence (since both policies are Gaussian in our implementation, a closed form of their KL exists). Such a closed form version for the Tsallis based Bregman is quite cumbersome to handle in terms of implementation, and thus we did not pursue the Tsallis based version in the on-policy experiments. However, in principle, it is very much feasible and we leave this for future investigation.

# D    REVERSE VS. FORWARD KL DIRECTION

Similar to the on-policy case, the *mode-seeking* or *reverse* direction of the KL term in off-policy MDPO (Eq. 16) is consistent with that in the MD update rule in convex optimization. With this direction of KL, the optimization problems for policy update in both off-policy MDPO and SAC are invariant to the normalization term $Z(s)$. Thus, these algorithms can update their policies without computing $Z(s)$. In [21], the authors proposed an algorithm, called *exploratory conservative policy optimization* (ECPO), that resembles our *soft* off-policy MDPO, except in the direction of KL. Switching the direction of KL to *mean-seeking* or *forward* has the extra overhead of estimating the normalization term for ECPO. However, in [21], they argue that it results in better performance. They empirically show that ECPO performs better than several algorithms, including one that is close to off-policy MDPO, which they refer to as *policy mirror descent* (PMD), and report poor performance for it. We did not use their code-base and exact configuration, but we did not observe such poor performance for our off-policy MDPO. In fact, experimental results of Section 5.4 show that off-policy MDPO performs better than or on-par with SAC in six commonly used MuJoCo domains. More experiments and further investigation are definitely required to better understand the effect of the KL direction in MDPO algorithms.

# E  ON-POLICY RESULTS

Here, we report the results for all on-policy algorithms, i.e. TRPO, PPO and MDPO. We have three variants here, **1)** the *minimal version*, i.e. {TRPO, PPO, MDPO}-M, which makes use of no code level optimizations, **2)** the *loaded version*, i.e. {TRPO, PPO, MDPO}-LOADED, which includes all code level optimizations, and **3)** the *loaded+GAE* version, i.e. {TRPO, PPO, MDPO}-LOADED+GAE, which includes all code level optimizations and also includes the use of GAE. We see that the overall performance increases in most cases as compared to the minimal versions. However, the trend in performance between these algorithms remains consistent to the main results.

|  | MDPO | TRPO | PPO |
|---|---|---|---|
| Hopper-v2 | 1964 $(\pm217)$ | **2382** $(\pm\mathbf{445})$ | 1281 $(\pm353)$ |
| Walker2d-v2 | **2948** $(\pm\mathbf{298})$ | 2454 $(\pm171)$ | 424 $(\pm92)$ |
| HalfCheetah-v2 | **2873** $(\pm\mathbf{835})$ | 1726 $(\pm690)$ | 617 $(\pm135)$ |
| Ant-v2 | 1162 $(\pm738)$ | **1716** $(\pm\mathbf{338})$ | -40 $(\pm33)$ |
| Humanoid-v2 | **635** $(\pm\mathbf{46})$ | 449 $(\pm9)$ | 448 $(\pm56)$ |
| HumanoidStandup-v2 | **127901** $(\pm\mathbf{6217})$ | 100408 $(\pm12564)$ | 96068 $(\pm11721)$ |

Table 5: Performance of MDPO-M, compared against PPO-M, TRPO-M on six MuJoCo tasks. The results are averaged over 5 runs, together with their 95% confidence intervals. The values with the best mean scores are bolded.

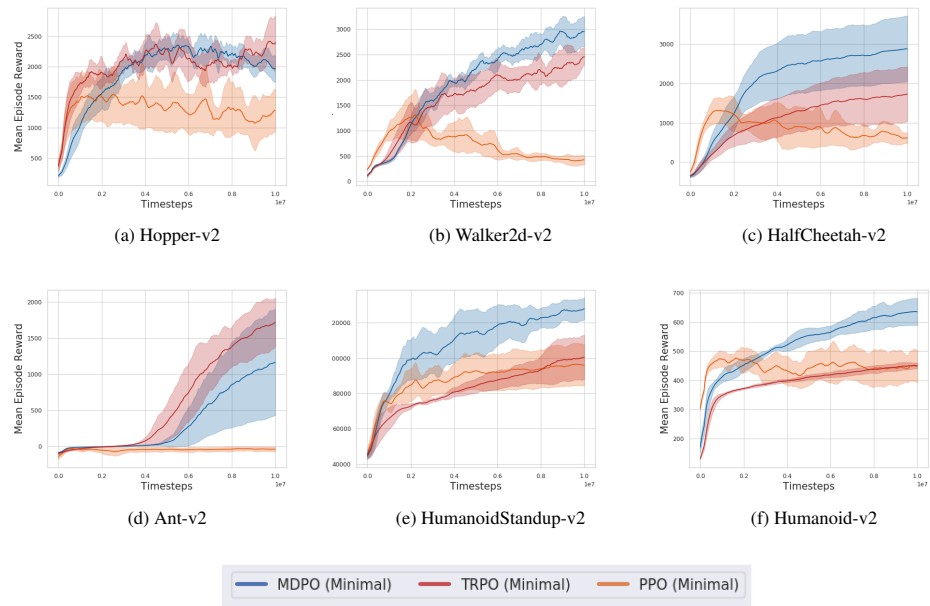

Figure 3: Performance of MDPO-M, compared against PPO-M, TRPO-M on six MuJoCo tasks. The results are averaged over 5 runs, with their 95% confidence intervals shaded.

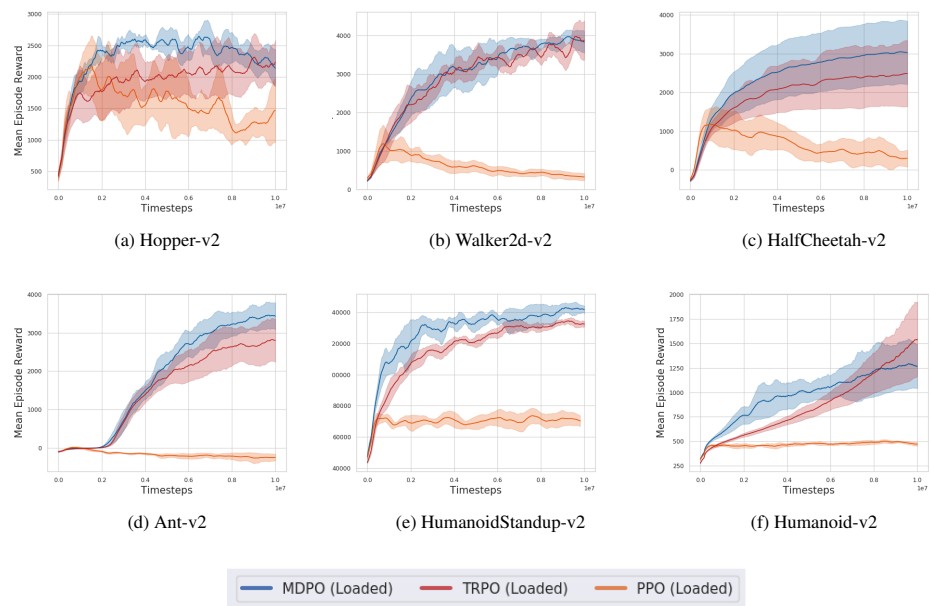

Figure 4: Performance of MDPO-LOADED, compared against loaded implementations (excluding GAE) of PPO and TRPO (PPO-LOADED, TRPO-LOADED) on six MuJoCo tasks. The results are averaged over 5 runs, with their 95% confidence intervals shaded.

|  | MDPO | TRPO | PPO |
|---|---|---|---|
| Hopper-v2 | **2361** (±**518**) | 1979 (±672) | 2051 (±241) |
| Walker2d-v2 | **4834** (±**607**) | 4473 (±558) | 1490 (±292) |
| HalfCheetah-v2 | **4172** (±**1156**) | 3751 (±910) | 2041 (±1319) |
| Ant-v2 | **5211** (±**43**) | 4682 (±278) | 59 (±133) |
| Humanoid-v2 | 3234 (±566) | **4414** (±**132**) | 529 (±47) |
| HumanoidStandup-v2 | **155261** (±**3898**) | 149847 (±2632) | 97223 (±4479) |

Table 6: Performance of MDPO-LOADED+GAE, compared against loaded implementations (including GAE) of PPO and TRPO (PPO-LOADED+GAE, TRPO-LOADED+GAE) on six MuJoCo tasks. The results are averaged over 5 runs, together with their 95% confidence intervals. The values with the best mean scores are bolded.

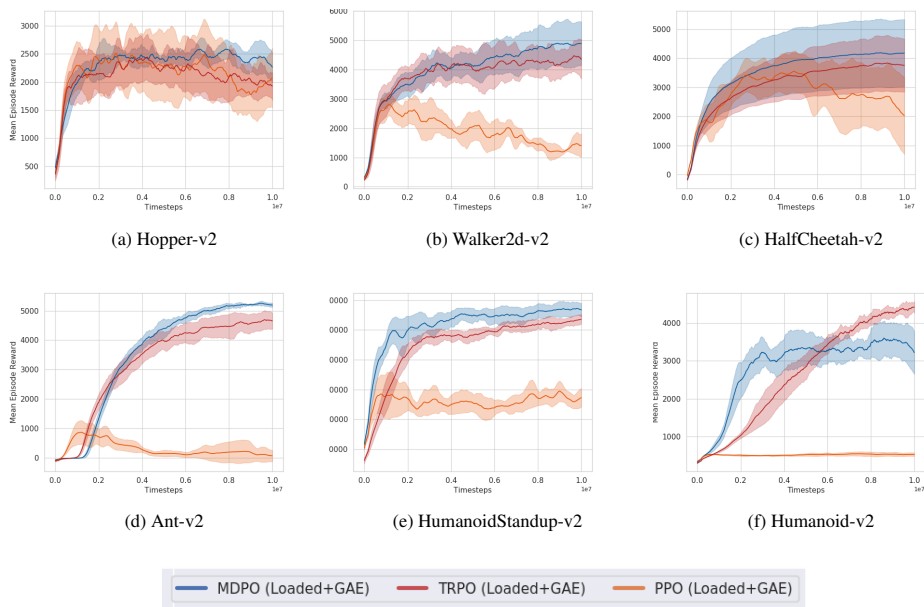

(a) Hopper-v2      (b) Walker2d-v2      (c) HalfCheetah-v2

(d) Ant-v2      (e) HumanoidStandup-v2      (f) Humanoid-v2

MDPO (Loaded+GAE)    TRPO (Loaded+GAE)    PPO (Loaded+GAE)

Figure 5: Performance of MDPO-LOADED+GAE, compared against loaded implementations (including GAE) of PPO and TRPO (PPO-LOADED+GAE, TRPO-LOADED+GAE) on six MuJoCo tasks. The results are averaged over 5 runs, with their 95% confidence intervals shaded.

# F  OFF-POLICY RESULTS

Here, we report the results for all off-policy algorithms, i.e. MDPO and SAC. We have two variants here, **1)** the *minimal version*, i.e. MDPO-M and SAC-M, which uses the standard neural network and batch sizes (64) and **2)** the *loaded version*, i.e. MDPO-LOADED and SAC-LOADED, which uses a neural network and batch size of 256. We see that the overall performance increases in most cases as compared to the minimal versions, i.e. SAC-M, MDPO-M. However, the trend in performance between these algorithms remains consistent to the main results.

|  | MDPO-KL | MDPO-Tsallis, $q_{\text{best}}$ | SAC |
|---|---|---|---|
| Hopper-v2 | 1385 $_{(\pm 648)}$ | 1385 $_{(\pm 648)}$, $q = 1.0$ | **1501** $_{(\pm 414)}$ |
| Walker2d-v2 | 873 $_{(\pm 180)}$ | **1151** $_{(\pm 218)}$, $q = 1.8$ | 635 $_{(\pm 137)}$ |
| HalfCheetah-v2 | 8098 $_{(\pm 428)}$ | 8477 $_{(\pm 450)}$, $q = 1.4$ | **9298** $_{(\pm 371)}$ |
| Ant-v2 | 1051 $_{(\pm 284)}$ | **2348** $_{(\pm 338)}$, $q = 2.0$ | 378 $_{(\pm 33)}$ |
| Humanoid-v2 | 2258 $_{(\pm 372)}$ | **4426** $_{(\pm 229)}$, $q = 1.6$ | 3598 $_{(\pm 172)}$ |
| HumanoidStandup-v2 | 131702 $_{(\pm 7203)}$ | 138157 $_{(\pm 8983)}$, $q = 1.2$ | **142774** $_{(\pm 4864)}$ |

Table 7: Performance of KL and Tsallis based versions of MDPO-M, compared with SAC-M on six MuJoCo tasks. The results are averaged over 5 runs, together with their 95% confidence intervals. The values with the best mean scores are bolded.

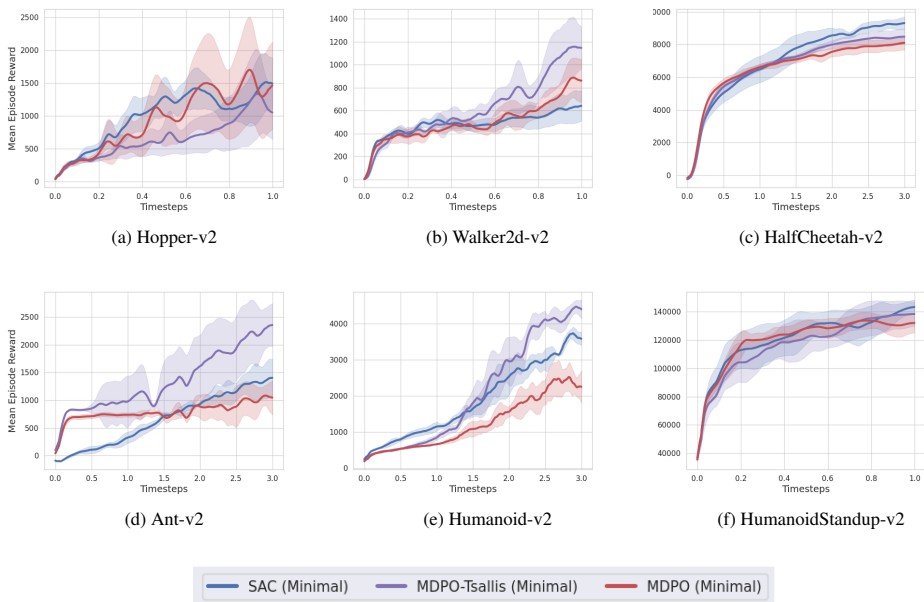

Figure 6: Performance of KL and Tsallis based versions of MDPO-M, compared with SAC-M on six MuJoCo tasks. X-axis represents time steps in millions. The results are averaged over 5 runs, with their 95% confidence intervals shaded.

|  | MDPO-KL | MDPO-Tsallis, $q_{\text{best}}$ | SAC |
|---|---|---|---|
| Hopper-v2 | **2428** ($\pm$**395**) | **2428** ($\pm$**395**), $q = 1.0$ | 1870 ($\pm$404) |
| Walker2d-v2 | 3591 ($\pm$366) | **4028** ($\pm$**287**), $q = 2.0$ | 3738 ($\pm$312) |
| HalfCheetah-v2 | 11823 ($\pm$154) | 11823 ($\pm$154), $q = 1.0$ | **11928** ($\pm$**342**) |
| Ant-v2 | 4434 ($\pm$749) | **5486** ($\pm$**737**), $q = 2.0$ | 4989 ($\pm$579) |
| Humanoid-v2 | 5323 ($\pm$348) | **5611** ($\pm$**260**), $q = 1.2$ | 5191 ($\pm$312) |
| HumanoidStandup-v2 | 143955 ($\pm$4499) | **165882** ($\pm$**16604**), $q = 1.4$ | 154765 ($\pm$11721) |

Table 8: Performance of KL and Tsallis based versions of MDPO-LOADED, compared with SAC-LOADED on six MuJoCo tasks. The results are averaged over 5 runs, together with their 95% confidence intervals. The values with the best mean scores are bolded.

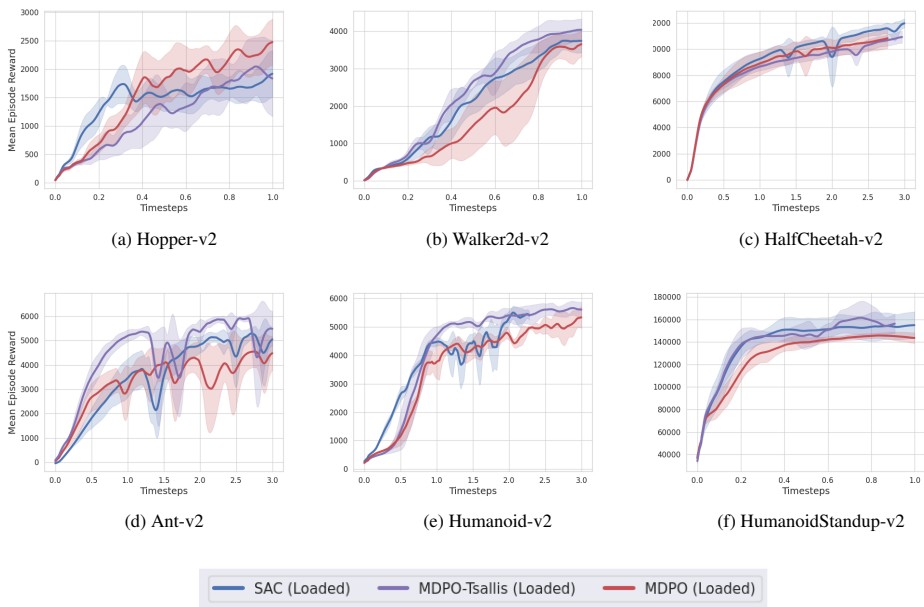

Figure 7: Performance of KL and Tsallis based versions of MDPO-LOADED, compared with SAC-LOADED on six MuJoCo tasks. X-axis represents time steps in millions. The results are averaged over 5 runs, with their 95% confidence intervals shaded. Note that although there is overlap in the performance of all methods, MDPO achieves a higher mean score in 5 out 6 domains.

## G  ADDITIONAL EXPERIMENTS

### G.1  MULTI-STEP UPDATE

For off-policy MDPO, we use a modified version of doing multi-step updates at each iteration (see section 5.1) due to computational reasons. In order to ensure a fair comparison, we used the single-step gradient updates for SAC in our main experiments. Here, we resort to the original algorithm presented in Algorithm 2, wherein we do $m$ gradient steps at each iteration. We compare this version of MDPO-KL with a similar multi-step version of SAC, as is originally reported in [13]. In Figure 8 we see that doing multiple updates helps improve the score of both MDPO and SAC, as is expected.

Figure 8: Performance of off-policy MDPO, compared with SAC on Hopper-v2, when doing both single and multiple gradient updates each iteration.

### G.2  DIFFERENT TSALLIS ENTROPIES FOR BREGMAN AND MDP REGULARIZATION

So far in the paper, we have used the same Tsallis entropy (same $q$ value) for defining the Bregman divergence as well as the MDP regularizer. Here, we test the performance for when the two tsallis entropies are different. For this, we sample from a set of three $q$ values $\{1.0, 1.5, 2.0\}$ and report the results for every possible combination of $q$ values used for defining the Bregman divergence and the MDP regularizer. We test this on the Walker2d-v2, Humanoid-v2, and Ant-v2 domains (domains where we see the most improvement due to the addition of Tsallis entropy) and observe that sticking to the same $q$ values for both cases results in the best performance across all three domains (see Table 9).

|  | MDP $q$ | Bregman $q$ | | |
|---|---|---|---|---|
|  |  | q = 1.0 | q = 1.5 | q = 2.0 |
| Walker2d-v2 | q = 1.0 | **3591** ($\pm$**366**) | 3268 ($\pm$234) | 1007 ($\pm$422) |
|  | q = 1.5 | 2126 ($\pm$456) | **2805** ($\pm$**302**) | 1573 ($\pm$328) |
|  | q = 2.0 | 14 ($\pm$5) | 2915 ($\pm$391) | **4028** ($\pm$**287**) |
| Ant-v2 | q = 1.0 | **4434** ($\pm$**749**) | 3007 ($\pm$572) | 1913 ($\pm$973) |
|  | q = 1.5 | 4119 ($\pm$326) | **5488** ($\pm$**233**) | 2781 ($\pm$812) |
|  | q = 2.0 | -807 ($\pm$951) | 4418 ($\pm$184) | **5486** ($\pm$**737**) |
| Humanoid-v2 | q = 1.0 | **5323** ($\pm$**348**) | 4734 ($\pm$341) | 4561 ($\pm$381) |
|  | q = 1.5 | 24 ($\pm$4) | **5013** ($\pm$**274**) | 3766 ($\pm$331) |
|  | q = 2.0 | 12 ($\pm$5) | 28($\pm$3) | **2751** ($\pm$**304**) |

Table 9: Different Tsallis Entropies. The results are averaged over 5 runs, with 95% confidence intervals shaded.

### G.3  TSALLIS-BASED SAC

We test performance of SAC-Tsallis while varying the $q$ values. In our preliminary experiments with SAC-Tsallis in Table 9, we did not see much improvement over SAC by tuning $q$, unlike what we observed in our MDPO-Tsallis results. More experiments and further investigation are definitely needed to better understand the effect of Tsallis entropy (and $q$) in these algorithms.

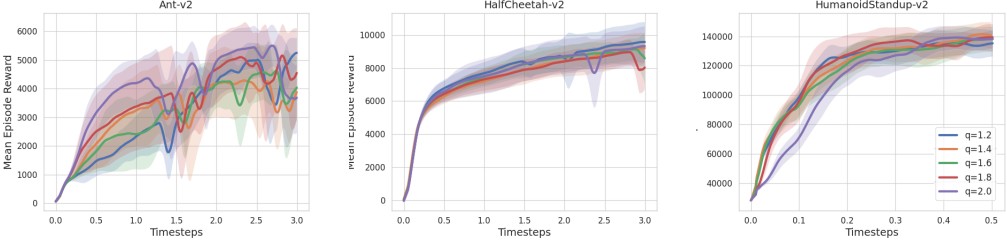

Figure 9: Performance of Tsallis SAC. The results are averaged over 5 runs, with 95% confidence intervals shaded.

# H  ATARI RESULTS

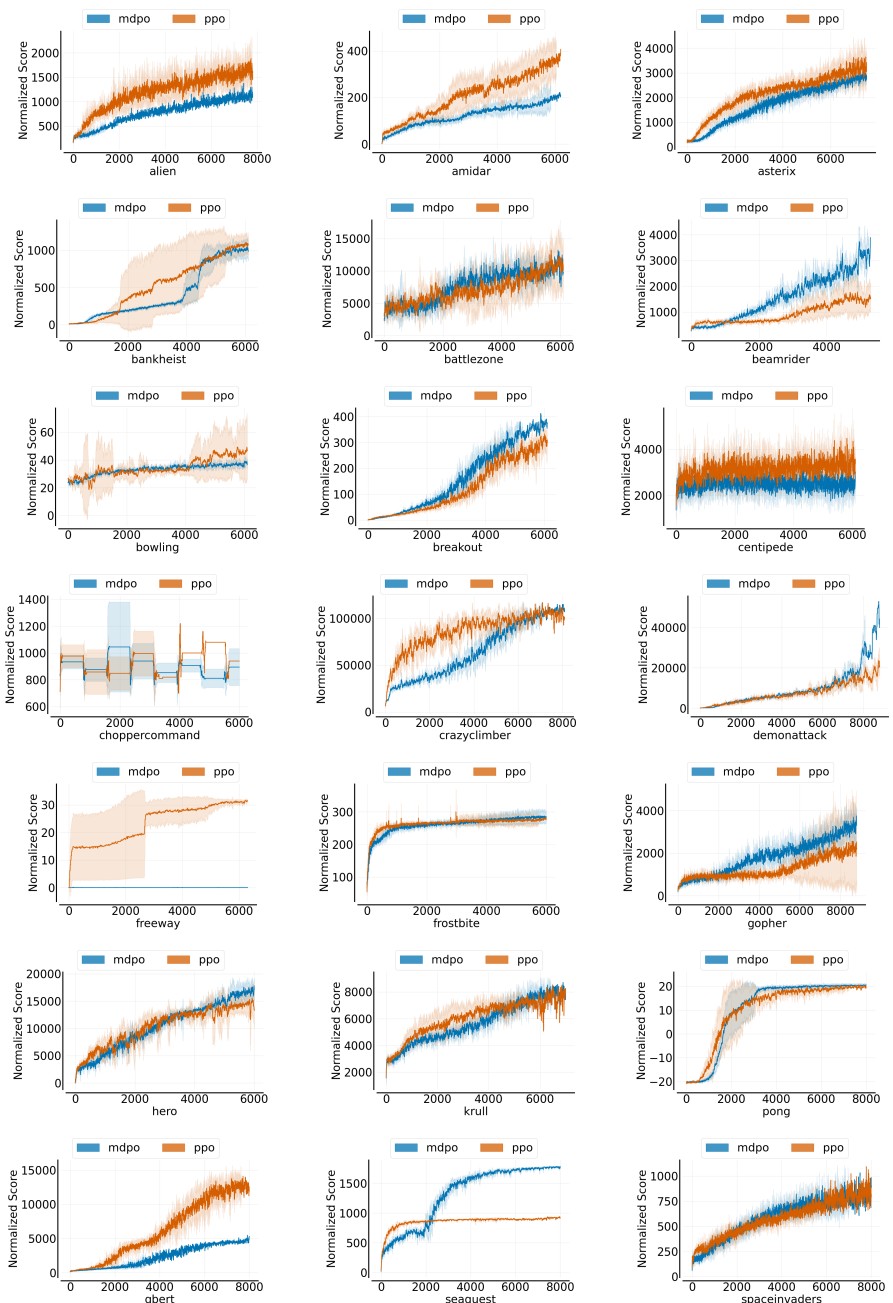

Figure 10: Comparison of MDPO with PPO on 21 Atari games. Order of magnitude of x-axis is $10^3$, which roughly corresponds to 10M environment time steps or 40M game frames.

