# OpenReview forum: "Mirror Descent Policy Optimization"
_ICLR.cc/2022/Conference — ICLR 2022 Poster_

### Official Review · Reviewer_qkZc · 2021-11-02

**Correctness:** 3
**Technical Novelty And Significance:** 1
**Empirical Novelty And Significance:** 2
**Recommendation:** 5
**Confidence:** 4

**Main Review:**

Strengths:
- The paper is well written, and the motivation is easy to follow.
- Going beyond the tabular RL, the paper derives a scalable algorithm from the MD principle. Moreover, the algorithm is practical since it allows for function approximation.
- The result is well supported empirically over several different tasks, which shows the advantage of MDPO.

Weakness:
- The paper does not provide any theoretical analysis for the algorithm. While the convergence and optimality of mirror descent are well-known for the convex constrained problems, it is still not sure if reinforcement learning lies in such a framework.
- My primary concern is about the novelty of the paper. The connection between PPO, TRPO, and mirror descent is well-known (see, e.g. [11, 19, 20, 24, 28] and [A]). Extending the tabular setting to the function approximation setting is also well studied in [19, 26. 27] both theoretically and empirically.

[A] Zhan, Wenhao, et al. "Policy mirror descent for regularized reinforcement learning: A generalized framework with linear convergence." arXiv preprint arXiv:2105.11066 (2021).


**Summary Of The Paper:**

The paper connects the optimization method, mirror descent, to the study of the policy optimization method. Based on the mirror descent principle, the paper proposes the MDPO algorithm, which updates the policy via approximately solving a trust-region problem. The paper proposes the on-policy and off-policy variants of MDPO. Furthermore, the paper connects the on-policy MDPO to PPO and TRPO and connects the off-policy MDPO to SAC. The contribution of the paper is to provide a unified viewpoint of several RL algorithms and shows that   MDPO performs equally or better than TRPO, PPO, and SAC in different tasks.


**Summary Of The Review:**

Overall, the paper is well written, and the empirical study well supports the result. However, the paper lacks the theoretical analysis, and the novelty of deriving MDPO from MD is limited. Given that, I recommend rejection for the paper.

---

> ### Author Response · Authors · 2021-11-14
> **Author Response**
>
> Hi, thank you for your efforts in reviewing our paper. Below, we try to address the raised concerns:
>
> With regards to novelty and including a theoretical analysis, although MD theory is well established for tabular RL, when moving to the function approximation case, one important question that arises is how to solve for the proximal step. Here, we present one possible idea to do this, something prior work like PPO and TRPO were missing. We believe that the role of any theory in the function approximation case should be to inform the practical algorithm. Specifically, while works such as [19] are useful since they complete missing results, they do not particularly help inform our practical algorithm. What helps the algorithm design is the general idea of doing MD over RL states, which is well captured in the papers that deal with the tabular setting as well. As pointed out by Reviewer fuGH, the focus of this paper is algorithmic and empirical in nature, and thus we do not believe that theoretical results are necessary for this paper to be published at a conference such as ICLR.
>
> Side note: The connection between MD and RL is well established in a lot of prior work, as you have noted through [11, 19, 20, 24, 28, [A]]. And so we are unsure of what you mean by “While the convergence and optimality of mirror descent are well-known for the convex constrained problems, it is still not sure if reinforcement learning lies in such a framework.“. Can you please clarify this?
>
> We hope our reply addresses your concerns and if so, you reconsider your assessment.

---

> > ### Comment · Reviewer_qkZc · 2021-11-29
> > **Reply**
> >
> > Thank you for your clarification. Now I see that the paper mainly focuses on algorithmic design and empirical study. However, my concern still comes from the novelty of the paper, which seems to be a combination of mirror descent, policy optimization, and function approximation. In my opinion, the paper lies below the borderline of acceptance.

---

### Official Review · Reviewer_wxw9 · 2021-11-02

**Correctness:** 3
**Technical Novelty And Significance:** 2
**Empirical Novelty And Significance:** 3
**Recommendation:** 6
**Confidence:** 2

**Main Review:**

1. The paper is well-written and easy to follow. More specifically, this work compared the novel algorithm with previous work (PPO, TRPO, SAC) carefully and clarified the experiment setting precisely.

2. The experiment parts contain some continuous control tasks, and this result highly supports the efficiency of the proposed algorithm compared with previous work (PPO, TRPO, SAC).

3. However, I still have some concerns about this work. It seems that this paper this work is a combination of mirror gradient descent with the previous TRPO or PPO structure, and the technical contribution is limited. Furthermore, the author compares the TRPO or PPO  algorithm, and it seems that none of the differences will improve the experiments. So it looks better if the author can make more discussion of the intuition compared with previous work.

**Summary Of The Paper:**

This paper focuses on reinforcement learning with the tabular Markov decision process setting and proposes the mirror descent policy optimization for off-policy and on-policy situations. Furthermore, the experiment result shows that this new algorithm outperforms the previous algorithm in both off-policy and on-policy situations.

**Summary Of The Review:**

Based on the previous comment, it is marginally above the acceptance threshold due to the excellent experiment result.

---

> ### Author Response · Authors · 2021-11-14
> **Author Response**
>
> Hi, thank you for your efforts in reviewing our paper and for your positive feedback. Below, we would like to address your concern regarding the technical contribution:
>
> Even while following the TRPO or PPO 'structure', we have shown that MDPO is actually quite different from the previously proposed algorithms. Particularly, TRPO is very different since it enforces a hard constraint while PPO uses a clipping objective. Importantly, MDPO is a straightforward implementation of the well understood MD theory in RL (unlike PPO), which also results in easier implementation and faster training. We also show that a simple annealing rule for the learning rate, inspired by the theory of mirror descent, is sufficient for the success of MDPO, unlike TRPO and PPO that use different heuristics that are harder to tune for their learning rates. Since MDPO offers the best of all algorithms in terms of performance, hyper-parameters, and wall clock times, while being grounded to theory, we believe the algorithm choices are useful contributions to the RL community.
>
> Side note: Can you please elaborate on " it seems that none of the differences will improve the experiments. So it looks better if the author can make more discussion of the intuition compared with previous work." We were not able to understand this as is.

---

> > ### Comment · Reviewer_wxw9 · 2021-11-30
> > **Keep my score**
> >
> > Thanks for your reply and I will keep my score unchanged.

---

### Official Review · Reviewer_fuGH · 2021-11-02

**Correctness:** 4
**Technical Novelty And Significance:** 3
**Empirical Novelty And Significance:** 3
**Recommendation:** 8
**Confidence:** 2

**Main Review:**

The paper proposed an algorithm inspired by Mirror descent, much like TRPO and PPO, both of which are well known and used.
The method appears to work convincingly better than the state of the art in the proposed experiments.
For the on policy setting, an interesting difference with TRPO is the way the trust region subproblem is solved.
A great strength of this work is that it nicely connects with the available literature, clearly summarizing the state of the art as well as the algorithmic differences. A reference that could provide some value is 'On the Theory of Policy Gradient Methods: Optimality, Approximation, and Distribution Shift' by Agarwal et al.

**Summary Of The Paper:**

The work proposes an algorithm like TRPO and PPO inspired by Mirror Descent. The contribution of the paper is primarily algorithmic and also empirical; there is no supporting theory for the proposed algorithm but I believe this is fine for this type of paper.


**Summary Of The Review:**

I support the paper for publication. I come from more from the theory community, and therefore ask the metareviewer to discount my ( unfortunately brief) review accordingly.

---

> ### Author Response · Authors · 2021-11-14
> **Author Response**
>
> Hi, thank you for your efforts in reviewing our paper and for the positive review. Please let us know if you have any other questions.
>
> We have added a reference to Agarwal et al. Thanks for noting this.

---

### Official Review · Reviewer_ELQd · 2021-11-03

**Correctness:** 4
**Technical Novelty And Significance:** 4
**Empirical Novelty And Significance:** 4
**Recommendation:** 6
**Confidence:** 4

**Main Review:**


Main Review:
Overall I think this could be a good paper. I am really interested in the connections between PPO, TRPO and SAC and MDPO seems to be a more general version of all three of them. Experimentally the results are very sound and seem to outperform existing methods while also being a simpler algorithm with fewer hyperparameters. I could see this method becoming very popular as a standard RL baseline.

With that being said, I think the off-policy section needs some work. Equation 11 is described in the appendix but it is not described when it is introduced. Can you include a description of how this equation works? At least give a reference to the appendix please. In general I thought the entire paragraph starting with “the main idea in algorithm 2…” was poorly written and confusing. Maybe this should be expanded into a couple of paragraphs. Can you explicitly write out the soft version of off-policy MDPO instead of just describing it in the text? Maybe include a new algorithm? “If we write the definition of KL and use the reparameterization trick in(16), we will rederive the loss function(11)used by our off-policy MDPO algorithm” I don’t think this is true, wouldn’t it be the same as (16) but without the current policy term? Maybe you can write out this equation.

Also, the related works could be explained better. The authors did a great job describing the differences between MDPO and TRPO/PPO/SAC, but there are many other related methods out there that should be referenced and talked about, for example Trust-PCL.

Minor points:
> “Since finding an optimal policy for an MDP involves solving a non-linear system of equations and the optimal policy may be deterministic (less explorative),”

Not sure if this explains why one might use an entropy regularizer.

> “Finally, we ran a similar experiment for TRPO which shows that performing multiple gradient stepsat each iteration of TRPO does not lead to any improvement. In fact our results showed that in somecases, it even leads to worse performance than when performing a single-step update”

Where are these results? Can you reference them in the text?

Could you put a vertical line in the results to separate on policy and off policy algorithms?
I was surprised to see MDPO not learn at all in the freeway atari experiment. Could you add some more commentary on the atari experiments? PPO seems to do much better against MDPO than the mujoco experiments.
Could you add hyperlinks to references and figures/algorithms?

**Summary Of The Paper:**

Summary:
Inspired by recent theoretical analysis of TRPO and PPO that use mirror descent, this paper proposes two new algorithms that directly minimize a mirror descent objective by taking multiple gradient steps. While similar to TRPO and PPO, MDPO is different in important ways, and happens to perform better in practice. The authors also propose a similar off-policy version of MDPO that happens to be closely related to SAC while also outperforming SAC experimentally.




**Summary Of The Review:**

I really like this paper but think the off-policy section can be re-written to be more coherent.

---

> ### Author Response · Authors · 2021-11-14
> **Author Response**
>
> Hi, thank you for your efforts in reviewing our paper and for the positive feedback. Based on your comments, we have updated the manuscript with the following changes:
>
> - We have updated the description of Eq. 11 and how it is derived from Eq. 10 as follows (We have split it into two paragraphs. The first describes the high level idea, while the second discusses the derivation):
>
> *"In particular, the first two terms here are obtained just by opening the KL, whereas the advantage estimate is replaced by a neural network estimate $Q_\psi$, which is learned from off-policy data in a $\text{TD}(0)$ fashion. Furthermore, solely as an implementation detail, another neural network $V_\phi$ is used in conjunction with $Q_\psi$, which is fit to the $Q_{\psi}$ estimate of the current policy. Finally, the policy loss also uses the reparameterization trick where $\widetilde{a}_{\theta} (\epsilon, s)$ is the action generated by sampling the $\epsilon$ noise from a zero-mean normal distribution $\mathcal N$."*
>
> - We explicitly write out the soft version of off-policy MDPO in Algorithm 3 (in Appendix). This is placed next to the hard version (Alg 2) and the differences are highlighted.
>
> - We have added the following discussion on Trust-PCL:
> *"Trust-PCL uses the path consistency idea along with entropy regularization and an additional term for remaining close to a past policy. In principle, this resembles the off-policy MDPO algorithm. However, Trust-PCL uses a multi-step consistency loss whereas off-policy MDPO uses single transitions. Moreover, besides different derivations, there remain implementation-level details between the two, as Trust-PCL only uses a $V$ network while off-policy MDPO uses a $Q$ function as well."*
>
> >If we write the definition of KL and use the reparameterization trick in(16), we will rederive the loss function(11)used by our off-policy MDPO algorithm” I don’t think this is true, wouldn’t it be the same as (16) but without the current policy term? Maybe you can write out this equation.
>
> - Here, we write out how Eq. 11 can be derived from Eq. 16
>
> $ \mathcal{L} (\theta, \theta_k) = \mathbb{E}_{s \sim \mathcal{D}} \Big[ \text{KL} \big(s; \pi, \pi_k \big) \exp(t_k Q^{k}) \Big] $
>
> $ = \mathbb{E}_{s \sim \mathcal{D}} \Big[ \sum_a \pi \log \pi  - \sum_a \pi \log ( \pi_k \exp(t_k Q^{k})) \Big] $
>
> $ = \mathbb{E}_{s \sim \mathcal{D}, \ a \sim \pi} \Big[ \log \pi  - \log ( \pi_k \exp(t_k Q^{k})) \Big] $
>
> $ = \mathbb{E}_{s \sim \mathcal{D}, \ a \sim \pi} \Big[ \log \pi  - \log \pi_k  - \log \exp(t_k Q^{k})) \Big] $
>
> $ = \mathbb{E}_{s \sim \mathcal{D}, \ a \sim \pi} \Big[ \log \pi  - \log \pi_k  - t_k Q^{k})) \Big] $
>
> >Where are these results? Can you reference them in the text?
>
> - These were preliminary experiments we were trying with TRPO so we don’t have the exact numbers for these. We’ve rewritten the text to highlight this.
>
> >I was surprised to see MDPO not learn at all in the freeway atari experiment. Could you add some more commentary on the atari experiments? PPO seems to do much better against MDPO than the mujoco experiments.
>
> - Indeed, the Freeway experiment seems interesting. When looking into the critic and actor loss logs, we observed that the ppo policy is able to receive a slight positive reward very early which stops the critic from diverging. Whereas this does not happen for the MDPO policy. Note that this is in the first 100 steps, which is very less for the critic or the policy to shape up accurately. Therefore, it seems this is an exploration issue, where the PPO policy’s initialization is able to give it this slight boost in better exploration which snowballs into a slowly improving policy as training proceeds. On the Atari experiments, we were intuitively aware of this phenomenon that policy gradient methods can lead to quite diverse performance between continuous and discrete actions spaces. Indeed, we see that the discrete action space of Atari results in PPO performing in a much more stable fashion as compared to the continuous space of Mujoco.
>
> >Could you add hyperlinks to references and figures/algorithms?
>
> - Apologies! We realize how frustrating it can be to look across the math without hyperlinks. We didn’t notice that ICLR allowed appendices in the same PDF this year and split our paper which broke the links. We have fixed this in the updated PDF.
>
> We hope this addresses the reviewer's concerns/questions and if so, they reconsider their assessment.

---

### Public Comment · ~Yingru_Li1 · 2022-03-11
**Relevant related work**

Dear Authors,

We just came across this nice paper. You may find this paper from NeurIPS 2019 relevant: https://papers.nips.cc/paper/2019/hash/cc9657884708170e160c8372d92f3535-Abstract.html

---

### Decision · Program_Chairs · 2022-01-20

**Decision:**

Accept (Poster)

**Comment:**

This paper proposes and studies a variant of policy optimization---mirror descent policy optimization (MDPO)---which was inspired by the mirror descent algorithm in the optimization literature. The proposed algorithm attempts to find a policy parameter that maximizes the expected regularized advantage function,  where the regularization term is based on the KL divergence between the new policy iterate and the current policy iterate. The main contributions are algorithmic and empirical, with detailed discussions provided to illuminate the connection between MDPO and other existing policy optimization paradigms like TRPO, PPO, etc. The paper provides an interesting and useful contribution to the growing literature of policy optimization.